# A New Perspective on Hydrogen Chloride Scavenging at High Temperatures for Reducing the Smoke Acidity of PVC Cables in Fires V: Comparison between EN 60754-1 and EN 60754-2

**Iacopo Bassi [1], Claudia Bandinelli [1], Francesca Delchiaro [1], Marco Piana [2] and Gianluca Sarti [1,\*]**

1   Reagens S.p.A., Via Codronchi, 4, 40016 San Giorgio di Piano, Italy; iacopo.bassi@reagens-group.com (I.B.); claudia.bandinelli@reagens-group.com (C.B.); francesca.delchiaro@reagens-group.com (F.D.)
2   PVC Forum Italia, Via Giovanni da Procida, 11, 20149 Milano, Italy; marcopiana@pvcforum.it
\*   Correspondence: gianluca.sarti@fastwebnet.it

**Abstract:** Regulation (EU) No 305/2011 lays down harmonized conditions for marketing construction products in the European Union. One of its consequences has been the introduction of the product standard EN 50575 and standard EN 130501-6, concerning power, control, and communication cables permanently installed in buildings to prevent the risk of a fire and its consequences. EN 13501-6 provides the reaction to fire classifications for cables, the test methods to be performed, the requirements to meet a specific reaction to fire, and additional classifications for smoke production, flaming droplets, and acidity. It requires EN 60754-2 as the technical standard to assess acidity, and it defines three classes: $a_1$, $a_2$, and $a_3$ (the less performant). Due to the release of hydrogen chloride during the combustion, acidity is the weak point of PVC cables, which are not yet capable of achieving the $a_1$ or $a_2$ classes required for specific locations according to fire risk assessments. EN 13501-6 does not include EN 60754-1, used in harmonized standards outside the scope of Regulation (EU) No 305/2011. EN 60754-1 and EN 60754-2 are common standards for determining halogen gas content, and acidity/conductivity, respectively. While they involve the same type of test apparatus, they differ in heating regimes, final temperatures, and detection methods. In particular, EN 60754-2 requires testing at temperatures between 935–965 °C in the tube furnace, where the sample burns, the smoke is collected in bubblers, and pH and conductivity are measured as an indirect assessment of acidity. On the other hand, the temperature regime of EN 60754-1 is a gradual heating run, followed by isothermal heating at 800 °C. The paper shows that when potent acid scavengers are used in PVC compounds, performing EN 60754-2 with the thermal profile of EN 60754-1 or at 500 °C in isothermal conditions, the evolution of hydrogen chloride changes significantly up to 10 times less than the test performed in isothermal at 950 °C. The reason lies behind the kinetic of hydrogen chloride release during the combustion of PVC compounds: the higher the temperature or faster the heat release, the quicker hydrogen chloride evolution and the lower the probability for the acid scavenger to trap it. Thus, these findings emphasize the "fragility" of EN 60754-2 as a tool for assessing risks associated with the release of hydrogen chloride during fires.

**Keywords:** acid scavengers; PVC; cables; smoke acidity; acidity; construction product regulation

## 1. Introduction

### 1.1. The Additional Classification for Acidity and the Role of Acid Scavengers

In the European Union (EU), cables permanently installed in buildings must also be classified for acidity, and current PVC cables cannot meet the most stringent classifications required in some locations. This paper shows how acid scavengers work at different temperatures and temperature regimes and how they impact technical standards based on tube furnaces for assessing the hydrogen chloride (HCl) released in the gas phase during the combustion of PVC compounds for cables. These kinds of research play a crucial role

in the competition with halogen-free cables, intending to protect and improve the market of PVC compounds for cables.

### 1.2. The Background. Regulatory Context and Test Methods for Assessing the Acidity

The regulatory framework of building and construction works is laid down by the Regulation (EU) No 305/2011 (Construction Products Regulation, or CPR) that sets harmonized rules for evaluating the requirements of items permanently installed in residential and public buildings, considering the impacts on the environment and people's health and safety [1]. According to CPR, one of the basic requirements of construction works is safety in case of fire. In this context, a harmonized classification regarding reaction to fire and additional classifications for smoke production, flaming droplets, and acidity have been adopted [2–4]. Flooring, linear insulation for pipes, panels, wall coverings, and other items commonly found in buildings do not require tests and requirements for assessing the release of acid gases in case of fire [2]. However, cables are the only building and construction products for which an additional classification for acidity is needed [3].

EN 13501-6 provides the test methods and requirements for evaluating the reaction to fire classification of cables and their additional classifications [3]. EN 60754-2, originally developed to determine the corrosivity, is the technical standard used for assessing the acidity according to CPR [5], with the methodology also explained in detail in [6,7]. It is carried out by burning 1 g of test specimen in a tube furnace. The effluents are then collected in two bubblers with double deionized water (DDW), where pH and conductivity are measured. Weighted pH and conductivity values for the cable are calculated considering the non-metallic material per unit length of the cable, according to paragraph 8.3 of the standard [5]. Class $a_1$ cable requires the pH to be more than 4.3 and the conductivity less than 2.5 $\mu$S/mm, class $a_2$ requires the pH to be more than 4.3, and the conductivity less than 10 $\mu$S/mm and class $a_3$ are those materials that are neither class $a_1$ nor class $a_2$.

EN 60754-2 is performed under isothermal conditions between 935 °C and 965 °C. On the other hand, EN 60754-1, the technical standard performed for determining the halogen acid gas content in contexts outside CPR [8], is carried out with this heating regime: 40 min from room temperature to 800 °C and 20 min in isothermal conditions at 800 °C. In EN 60754-1, temperature increases at about 20 °C/min, covering the typical temperatures of ignition and the developing stage of the fire, then exceeding the typical flashover temperatures (600–650 °C) [9], and finally reaching 800 °C.

It should be highlighted that EN 60754-2 and EN 60754-1 are bench-scale tests. They do not consider all variables of a real fire scenario that could affect the concentration of HCl in the gas phase.

### 1.3. Fire Hazards: The Role of Flame Retardants, Smoke Suppressants, and Acid Scavengers

Flame retardants and smoke suppressants are crucial in designing PVC compounds for cables capable of delaying flashover, and decreasing smoke production, to meet the main goals of fire safety strategy, i.e., the reduction of fatalities and injuries, conservation of property, protection of environment, preservation of heritage, and continuity of business operations in case of fire. For evaluating the performance of flame retardants and smoke suppressants, different external heat fluxes can be chosen in bench-scale fire tests [9–11], to understand how to reduce the fire hazard, strictly linked to parameters like ignitability, flammability, heat release (amount and rate), flame spread, smoke production, and its toxicity [12]. For example, in cone calorimetry, flame retardants and smoke suppressants are usually tested in external heat fluxes typical of pre-flashover stages of a fire [9]. Among the parameters mentioned above, the heat release rate is considered "the single most important variable in fire hazard" [13], while the toxicity of the effluents is dominated by carbon monoxide (CO) and HCl plays a minor role in fire safety [14].

When a PVC cable burns, HCl is released from the polymer's thermal decomposition; therefore, it is one of the effluents in case of fire. However, in a real fire scenario, its concentration in the gas phase decays, absorbed by common materials found in buildings [14].

This behavior has two consequences: the HCl concentration in the gas phase is less than expected, and HCl does not travel far from where the fire originates. The presence of acid scavengers at high temperatures in PVC compounds for cables can decrease further the concentration of HCl in the gas phase in case of fire, but their use can interfere with flame retardants and smoke suppressants' action, as explained in [15].

*1.4. Acid Scavengers, Their Behavior at Different Temperatures and Novel Low Smoke Acidity PVC Compounds for Cables*

Current PVC cables in the market can only meet class $a_3$. Thus, the research in new low-smoke acidity compounds is paramount for making PVC cables, working in specific locations where the best additional classifications for acidity are needed [6,16]. A new generation of low-smoke acidity PVC compounds has been recently developed with the aim of manufacturing cables to meet the best additional classifications for acidity [17–19]. Despite improving the acidity, they failed to meet the class $a_1$ or $a_2$. These compounds contain acid scavengers that act at high temperatures in the condensed phase, efficiently trapping HCl in the char and reducing its evolution in the gas phase.

However, the acid scavengers' efficiency drops significantly at temperatures over 900 °C, according to the theory and data exhaustively reported in Parts I and II of this article [6,7]. In fact, at 950 °C, trivial coated ground calcium carbonate (GCC) has more or less the same performances as the more performant acid scavenger used in that research. In summary, their performance in dropping down the smoke acidity is strictly linked to the efficiency of the acid scavengers, and the efficiency depends on the kinetics of scavenging and whether acid scavengers or their reaction products are stable in the range of temperatures where we need them to work. Therefore, as described in [6], temperatures, heating regimes, and the chemical nature of the acid scavengers play a crucial role in their efficiency. Also, the acid scavengers' particle size and the dispersion level they can reach in the matrix, getting as much as possible intimate contact with PVC chains is paramount. Hence, the research in novel low-smoke acidity compounds at the laboratory level will be decisive, but the right choice of production systems able to reach a high dispersion level for acid scavengers will also be crucial. Cable manufacturers willing to do research aiming to produce PVC cables in class $a_1$ or $a_2$ must consider all these variables.

*1.5. The Impact of Acid Scavengers when EN 60754-2 Is Run at Different Thermal Profiles*

EN 60754-2 is run isothermally at temperatures between 935 °C and 965 °C. It is interesting and never shown before to see what happens when it is performed with different heating regimes in the presence and absence of acid scavengers to see their impacts when gradual heating runs are used or when isothermal temperatures below the flashover are applied. That emphasizes if the acid scavengers can show good efficiency at the temperatures typical of the stages of fires when a safe escape is possible.

*1.6. The Aim of the Research*

In this article, the acidity of several PVC compounds for cables has been tested by comparing the following test methods:

(1)　EN 60754-2 has been carried out isothermally at 950 °C, and EN 60754-2 with the heating regime of EN 60754-1 (internal method 3).

(2)　EN 60754-2 has been conducted at isothermally 950 °C, and EN 60754-2 isothermally at 500 °C (internal method 2).

The heating regime of EN 60754-1 is as follows: 40 min to 800 °C from room temperature and further heating of 20 min at 800 °C.

The aim has been to verify if different heating regimes could affect the concentration of HCl in the gas phase and to explore the role of the HCl scavengers in this context. In particular, it has been evaluated whether the acidity is reduced when the heating regime of EN 60754-1 is run and a pre-flashover temperature of 500 °C is chosen.

The high temperatures acid scavengers used in this research act in the condensed phase.

In this paper, "acidity" and "smoke acidity" are considered interchangeable terms.

The subsequent sections of this paper present the experimental material and methods, results, and discussion, followed by the conclusion and implications.

## 2. Materials and Methods

### 2.1. Materials

The tested formulations have been divided into four series to verify if the internal methods 2 and 3 give different results from EN 60754-2. Table 1 displays the first series and intends to show the impact of different acid scavengers at high temperatures on acidity, performing EN 60754-2, internal methods 2 and 3. FR50.0 is a typical PVC compound for non-flame retarded jackets, with a coated GCC with a low HCl scavenging efficiency like Riochim [20]. FR50.1 contains a synthetic $Al(OH)_3$ (ATH, from Nabaltec, Schwandorf, Germany), an inert acid scavenger, which does not reduce smoke acidity. FR50.2 has $Mg(OH)_2$ (MDH, from Europiren, Schiedam, The Netherlands) as uncoated brucite, an ineffective acid scavenger fixing HCl as $MgCl_2$ but then rereleasing it due to its decomposition [6,7,21,22]. FR50.3 includes coated ultrafine precipitated calcium carbonate (UPCC), Winnofil S from Imerys [23], a potent acid scavenger which efficiently captures HCl as $CaCl_2$ in a single-step reaction, currently used for reducing the acidity of the PVC compounds' effluents in case of fire [6,7]. FR50.4 and FR50.5 show the action of two potent acid scavengers from Reagens, AS-1B and AS-6B. They are the new generation of acid scavengers at high temperatures, acting in the condensed phase.

**Table 1.** First series of formulations: DINP means Di Iso Nonyl Phthalate. ESBO stands for Epoxidized Soy Bean Oil. The used antioxidant is Arenox A10, which is Pentaerythritol tetrakis(3-(3,5-di-tert-butyl-4-hydroxyphenyl)propionate), CAS number 6683-19-8. COS stands for Calcium Organic Stabilizer. UPCC means Ultrafine Precipitated Calcium Carbonate. HTAS stands for High Temperature Acid Scavengers.

| Raw Materials | Trade Name | FR50.0 [phr] | FR50.1 [phr] | FR50.2 [phr] | FR50.3 [phr] | FR50.4 [phr] | FR50.5 [phr] |
|---|---|---|---|---|---|---|---|
| PVC | Inovyn 271 PC | 100 | 100 | 100 | 100 | 100 | 100 |
| DINP | Diplast N | 50 | 50 | 50 | 50 | 50 | 50 |
| ESBO | Reaflex EP/6 | 2 | 2 | 2 | 2 | 2 | 2 |
| Antioxidant | Arenox A10 | 0.1 | 0.1 | 0.1 | 0.1 | 0.1 | 0.1 |
| COS | RPK B-CV/3037 | 3 | 3 | 3 | 3 | 3 | 3 |
| $CaCO_3$ | Riochim | 90 | 0 | 0 | 0 | 0 | 0 |
| $Al(OH)_3$ | Apyral 40 CD | 0 | 90 | 0 | 0 | 0 | 0 |
| $Mg(OH)_2$ | Ecopyren 3.5 | 0 | 0 | 90 | 0 | 0 | 0 |
| UPCC | Winnofil S | 0 | 0 | 0 | 90 | 0 | 0 |
| HTAS 1 | AS-1B | 0 | 0 | 0 | 0 | 90 | 0 |
| HTAS 2 | AS-6B | 0 | 0 | 0 | 0 | 0 | 90 |

Tables 2–4 show typical formulations with low values of acidity, where acid scavengers work in multiple-step reactions in fixing HCl in the condensed phase. These kinds of reactions are explained in [6,7]. The primary and secondary acid scavengers are dosed in different ratios giving different scavenging efficiencies, and the impacts on the measurements carried out with EN 60754-2 and internal 3 are evaluated. RPK B-NT/8014 is an anti-pinking additive from Reagens commonly used to switch off discoloration when large quantities of MDH are introduced in PVC compounds. AS0-B is a potent acid scavenger produced by Reagens. Cabosil H5 is a fumed silica from Cabot, RI004 antimony trioxide from Quimialmel, and Kisuma 5A, a synthetic coated MDH produced by Kisuma.

**Table 2.** Second series of formulations: DINP means Di Iso Nonyl Phthalate. ESBO stands for Epoxidized Soy Bean Oil. The used antioxidant is Arenox A10, which is Pentaerythritol tetrakis(3-(3,5-di-tert-butyl-4-hydroxyphenyl)propionate), CAS number 6683-19-8. COS stands for Calcium Organic Stabilizer. HTAS means High Temperature Acid Scavengers.

| Raw Materials | Trade Name | FR50.6 [phr] | FR50.7 [phr] | FR50.8 [phr] | FR950.9 [phr] |
|---|---|---|---|---|---|
| PVC | Inovyn 271 PC | 100.0 | 100.0 | 100.0 | 100.0 |
| DINP | Diplast N | 50.0 | 50.0 | 50.0 | 50.0 |
| ESBO | Reaflex EP/6 | 2.0 | 2.0 | 2.0 | 2.0 |
| Mg(OH)$_2$ | Kisuma 5A | 30.0 | 30.0 | 30.0 | 30.0 |
| Antioxidant | Arenox A10 | 0.1 | 0.1 | 0.1 | 0.1 |
| COS | RPK B-CV/3037 | 3.0 | 3.0 | 3.0 | 3.0 |
| HTAS 1 | AS-1B | 123.0 | 123.0 | 0.0 | 0.0 |
| HTAS 2 | AS-6B | 0.0 | 0.0 | 123.0 | 123.0 |
| Anti Pinking | RPK B-NT/8014 | 0.0 | 6.0 | 0.0 | 6.0 |

**Table 3.** Third series of formulations: DINP means Di Iso Nonyl Phthalate. ESBO stands for Epoxidized Soy Bean Oil. The used antioxidant is Arenox A10, which is Pentaerythritol tetrakis(3-(3,5-di-tert-butyl-4-hydroxyphenyl)propionate), CAS number 6683-19-8. COS stands for Calcium Organic Stabilizer. UPCC means Ultrafine Precipitated Calcium Carbonate. HTAS High Temperature Acid Scavenger.

| Raw Materials | Trade Name | FR50.10 [phr] | FR50.11 [phr] | FR50.12 [phr] | FR950.13 [phr] |
|---|---|---|---|---|---|
| PVC | Inovyn 271 PC | 100.0 | 100.0 | 100.0 | 100.0 |
| DINP | Diplast N | 50.0 | 50.0 | 50.0 | 50.0 |
| ESBO | Reaflex EP/6 | 2.0 | 2.0 | 2.0 | 2.0 |
| Antioxidant | Arenox A10 | 0.1 | 0.1 | 0.1 | 0.1 |
| COS | RPK B-CV/3037 | 3.0 | 3.0 | 3.0 | 3.0 |
| Mg(OH)$_2$ | Kisuma 5A | 30.0 | 30.0 | 30.0 | 30.0 |
| UPCC | Winnofil S | 90.0 | 90.0 | 0.0 | 0.0 |
| Fumed Silica | Cabosil H5 | 0.0 | 15.0 | 0.0 | 15.0 |
| HTAS 3 | AS-0B | 0.0 | 0.0 | 123.0 | 123.0 |

**Table 4.** Forth series of formulations: DINP means Di Iso Nonyl Phthalate. ESBO stands for Epoxidized Soy Bean Oil. The used antioxidant is Arenox A10, which is Pentaerythritol tetrakis(3-(3,5-di-tert-butyl-4-hydroxyphenyl)propionate), CAS number 6683-19-8. COS stands for Calcium Organic Stabilizer. ATO means antimony trioxide and HTAS High Temperature Acid Scavenger.

| Raw Materials | Trade Name | FR50.14 [phr] | FR50.15 [phr] | FR50.16 [phr] | FR50.17 [phr] |
|---|---|---|---|---|---|
| PVC | Inovyn 271 PC | 100.0 | 100.0 | 100.0 | 100.0 |
| DINP | Diplast N | 50.0 | 50.0 | 50.0 | 50.0 |
| ESBO | Reaflex EP/6 | 2.0 | 2.0 | 2.0 | 2.0 |
| Mg(OH)$_2$ | Ecopyren 3.5 | 30.0 | 30.0 | 30.0 | 30.0 |
| Antioxidant | Arenox A10 | 0.1 | 0.1 | 0.1 | 0.1 |
| COS | RPK B-CV/3037 | 3.0 | 3.0 | 3.0 | 3.0 |
| ATO | RI004 | 10.0 | 10.0 | 10.0 | 10.0 |
| CaCO$_3$ | Riochim | 0.0 | 0.0 | 0.0 | 65.0 |
| HTAS 1 | AS-1B | 123.0 | 0.0 | 0.0 | 0.0 |
| HTAS 2 | AS-6B | 0.0 | 123.0 | 0.0 | 0.0 |
| HTAS 3 | AS-0B | 0.0 | 0.0 | 123.0 | 0.0 |

EN 60754-2 and internal methods 2 and 3 use DDW internally produced by the ion exchange deionizer in Table 5 with the quality according to the standard (pH between

5.50 and 7.50, and conductivity less than 0.5 µS/mm). Buffer and conductivity standard solutions from VWR International are the following:

-   pH: 2.00, 4.01, 7.00, and 10.00,
-   conductivity: 2.0, 8.4, 14.7, 141.3 µS/mm

### 2.2. Test Apparatuses

Table 5 gives the employed test apparatuses.

**Table 5.** Main test apparatuses utilized.

| Test Apparatus | Producer | Model | Additional Info's |
|---|---|---|---|
| Torque Rheometer | Brabender | Plastograph EC | 50 CC chamber, 30 rpm, 60 g sample mass, 160 °C per 10 min. |
| Halogen Acid Gas test apparatus | SA Associates | Standard model | Porcelain combustion boats. |
| Multimeter | Mettler Toledo | S213 standard kit | |
| Conductivity electrode | Mettler Toledo | S213 standard kit | Reference thermocouple adjusting temperature fluctuation. |
| pH electrode | Mettler Toledo | S213 standard kit | Reference thermocouple adjusting temperature fluctuation. |
| Ion Exchange Deionizer | Culligan Pharma | System 20 | |

### 2.3. Sample Preparation

The formulations in Tables 1–4 were prepared in a turbo mixer, making the dry blends and then processing them in a torque rheometer. Dry blends were produced as follows. PVC and all additives, except plasticizer, were mixed up to 80 °C. Then plasticizer was added slowly until its complete "absorption." When the temperature reached 105 °C, the dry blend was unloaded in PE bags and left there for a full "maturation" of 24 h, then, 60 g of the dry blend were processed in the torque rheometer for 10 min Appendix A in Figures A1 and A2 details sample preparation, test apparatuses, and conditions.

The test specimens for EN 60754-2 and internal methods 2 and 3 have been derived from the kneaders.

### 2.4. Internal Tests and International Technical Standards Used

Table 6 shows the technical standards and the main utilized conditions.

**Table 6.** Tests for assessing acidity.

| Technical Standard | Measurement | Temperature | Note |
|---|---|---|---|
| EN 60754-2 | pH and conductivity | Isothermal at 950 °C | The general method, according to the 2014 version. |
| Internal method 2 | pH and conductivity | Isothermal at 500 °C | The general method, according to the 2014 version. |
| Internal method 3 | pH and conductivity | 23–800 °C in 40 min 800 °C per 20 min | EN 60754-2 carried out with the thermal profile of EN 60754-1 |

The procedures and the precautions in performing EN 60754-2 and internal methods 2 are described in detail in the technical standard [5], and Parts I and II of this paper [6,7].

Internal method 3 follows this specific procedure: an empty combustion boat is introduced into the tube furnace through the sample carrier. The airflow is set between 290 and 310 mL/min, depending on the quartz tube geometry. The thermocouple is placed at the center of the tube furnace, the initial ramp is chosen, the heater is started, and the time is measured with a stopwatch. The ramp is selected to reach $800 \pm 10$ °C in $40 \pm 5$ min and to maintain an isothermal condition of $800 \pm 10$ °C for $20 \pm 1$ min. The heating rate is adjusted accordingly if temperatures and times exceed the above ranges. The conductivity of the water in the bubblers is checked to verify the possibility of contamination from previous tests. After determining the quartz tube's heating regime and cleaning status, the sample is weighed in the combustion boat ($1.000 \pm 0.001$ g of material) and introduced into the tube furnace at room temperature through the sample carrier. The

heater is switched on, and the stopwatch monitors the ramp. After 1 h, the connectors are opened, the water from the bubbling devices and washing procedures is collected in a 1 L volumetric flask filled to the mark, and then pH and conductivity are measured as follows. The multimeter is calibrated with standard solutions before each measurement. The pH is calibrated at two points (4.01 and 7.00) while conductivity is at 1 point at 141.3 µS/mm. Then pH and conductivity of the quotes from the flask are measured. The solutions closer to the measured values are used as correction standards, and the measurements are corrected accordingly. pH and conductivity electrodes have a reference thermocouple that adjusts the temperature fluctuation. The method measures three replicates to calculate the mean value, standard deviation (SD), and coefficient of variation (CV).

Appendix A, Figures A1 and A2 provide a schematic diagram of the sample preparation, conditions, and testing process.

Supplementary Materials file provides additional information regarding materials (Table S1), test apparatuses (Table S2), and standards (Table S3) in Section S1, sample preparation in Section S2, and test procedures in Section S3.

## 3. Results

Tables 7–9 show pH and conductivity of the formulations in Table 1 measured according respectively to EN 60754-2 at 950 °C, internal methods 3 and 2.

**Table 7.** pH and conductivities of the formulation in Table 1, according to EN 60754-2 at 950 °C. The mean values, standard deviations, and coefficient of variation are reported.

| Formulation | FR50.0 | FR50.1 | FR50.2 | FR50.3 | FR50.4 | FR50.5 |
|---|---|---|---|---|---|---|
| **pH** | 2.62 | 2.27 | 2.27 | 2.74 | 2.89 | 2.79 |
| $SD_{pH}$ | 0.03 | 0.10 | 0.02 | 0.06 | 0.08 | 0.02 |
| $CV_{pH}$ [%] | 1.15 | 4.41 | 0.88 | 2.19 | 2.77 | 0.72 |
| **Conductivity [µS/mm]** | 97.3 | 221.5 | 224.3 | 74.0 | 70.1 | 70.1 |
| SDc | 3.7 | 8.4 | 3.1 | 1.6 | 0.7 | 2.0 |
| CVc [%] | 3.8 | 3.8 | 1.4 | 2.2 | 1.0 | 2.9 |

**Table 8.** pH and conductivity of the formulation in Table 1, according to internal method 3. The mean values, standard deviations, and coefficient of variation are reported.

| Formulation | FR50.0 | FR50.1 | FR50.2 | FR50.3 | FR50.4 | FR50.5 |
|---|---|---|---|---|---|---|
| **pH** | 2.51 | 2.29 | 2.28 | 3.32 | 3.56 | 3.29 |
| $SD_{pH}$ | 0.02 | 0.04 | 0.02 | 0.00 | 0.00 | 0.02 |
| $CV_{pH}$ [%] | 0.80 | 1.75 | 0.88 | 0.00 | 0.00 | 0.61 |
| **Conductivity [µS/mm]** | 135.7 | 224.7 | 228.0 | 25.5 | 11.6 | 22.8 |
| SDc | 4.4 | 6.1 | 1.5 | 0.7 | 0.2 | 0.1 |
| CVc [%] | 3.2 | 2.7 | 0.7 | 2.7 | 1.7 | 0.4 |

**Table 9.** pH and conductivity of the formulation in Table 1, according to internal method 2 at 500 °C. The mean values, standard deviations, and coefficient of variation are reported.

| Formulation | FR50.0 | FR50.1 | FR50.2 | FR50.3 | FR50.4 | FR50.5 |
|---|---|---|---|---|---|---|
| **pH at 500 °C** | 2.48 | 2.41 | 2.41 | 3.73 | 3.70 | 3.69 |
| $SD_{pH}$ | 0.04 | 0.03 | 0.09 | 0.10 | 0.15 | 0.13 |
| $CV_{pH}$ [%] | 1.61 | 1.24 | 3.73 | 2.68 | 4.05 | 3.52 |
| **Conductivity at 500 °C [µS/mm]** | 139.1 | 177.2 | 177.3 | 7.7 | 8.2 | 8.6 |
| SDc | 1.2 | 2.5 | 6.2 | 0.3 | 0.4 | 0.3 |
| CVc [%] | 0.9 | 1.4 | 3.5 | 3.9 | 4.9 | 3.5 |

Table 10 gives the pH and conductivity of the formulations in Table 2 measured according to EN 60754-2 at 950 °C. Table 11 pH and conductivity performing internal method 3.

**Table 10.** pH and conductivity of formulations in Table 2, according to EN 60754-2 at 950 °C. The mean values, standard deviations, and coefficient of variation are reported.

| Formulation | FR50.6 | FR50.7 | FR50.8 | FR50.9 |
|---|---|---|---|---|
| **pH** | 4.17 | 4.18 | 4.31 | 4.14 |
| $SD_{pH}$ | 0.08 | 0.11 | 0.07 | 0.17 |
| $CV_{pH}$ [%] | 1.92 | 2.63 | 1.62 | 4.11 |
| **Conductivity [μS/mm]** | 3.2 | 2.8 | 2.5 | 3.9 |
| SDc | 0.1 | 0.4 | 0.1 | 1.0 |
| CVc [%] | 3.1 | 14.3 | 4.0 | 25.6 |

**Table 11.** pH and conductivities of formulations in Table 2, according to internal method 3. The mean values, standard deviations, and coefficient of variation are reported.

| Formulation | FR50.6 | FR50.7 | FR50.8 | FR50.9 |
|---|---|---|---|---|
| **pH** | 4.29 | 4.46 | 4.73 | 4.44 |
| $SD_{pH}$ | 0.01 | 0.09 | 0.35 | 0.28 |
| $CV_{pH}$ [%] | 0.23 | 2.02 | 7.40 | 6.31 |
| **Conductivity [μS/mm]** | 1.8 | 1.8 | 1.3 | 2.3 |
| SDc | 0.0 | 0.4 | 0.3 | 0.7 |
| CVc [%] | 0.0 | 22.2 | 23.1 | 30.4 |

Table 12 brings the pH and conductivity of the formulations in Table 3 measured according to EN 60754-2 at 950 °C. Table 13 gives pH and conductivity obtained by performing internal method 3.

**Table 12.** pH and conductivity of formulations in Table 3, according to EN 60754-2 at 950 °C. The mean values, standard deviations, and coefficient of variation are reported.

| Formulation | FR50.10 | FR50.11 | FR50.12 | FR50.13 |
|---|---|---|---|---|
| **pH** | 3.29 | 3.12 | 3.65 | 3.69 |
| $SD_{pH}$ | 0.01 | 0.06 | 0.17 | 0.07 |
| $CV_{pH}$ [%] | 0.30 | 1.92 | 4.66 | 1.90 |
| **Conductivity [μS/mm]** | 24.2 | 34.3 | 11.0 | 8.1 |
| SDc | 2.1 | 2.3 | 3.9 | 2.1 |
| CVc [%] | 8.7 | 6.8 | 35.2 | 25.9 |

**Table 13.** pH and conductivity of formulations in Table 3, according to internal method 3. The mean values, standard deviations, and coefficient of variation are reported.

| Formulation | FR50.10 | FR50.11 | FR50.12 | FR50.13 |
|---|---|---|---|---|
| **pH** | 4.10 | 3.62 | 4.33 | 4.35 |
| $SD_{pH}$ | 0.04 | 0.06 | 0.07 | 0.08 |
| $CV_{pH}$ [%] | 0.98 | 1.66 | 1.62 | 1.84 |
| **Conductivity [μS/mm]** | 3.9 | 10.7 | 2.1 | 2.0 |
| SDc | 0.3 | 1.6 | 0.7 | 0.2 |
| CVc [%] | 7.7 | 15.0 | 33.3 | 10.0 |

Table 14 shows the pH and conductivity of the formulations in Table 4 measured according to EN 60754-2 at 950 °C. Table 15 displays the pH and conductivities, performing internal method 3.

**Table 14.** pH and conductivities of formulation in Table 4, according to EN 60754-2 at 950 °C. The mean values, standard deviations, and coefficient of variation are reported.

| Formulation | FR50.14 | FR50.15 | FR50.16 | FR50.17 |
|---|---|---|---|---|
| **pH** | 4.18 | 4.20 | 4.03 | 2.63 |
| $SD_{pH}$ | 0.08 | 0.09 | 0.08 | 0.10 |
| $CV_{pH}$ [%] | 1.91 | 2.14 | 1.99 | 3.80 |
| **Conductivity [$\mu$S/mm]** | 3.0 | 3.2 | 4.0 | 92.8 |
| SDc | 0.5 | 0.2 | 0.5 | 3.2 |
| CVc [%] | 16.7 | 6.3 | 12.5 | 3.4 |

**Table 15.** pH and conductivities of formulation in Table 4, according to internal method 3. The mean values, standard deviations, and coefficient of variation are reported.

| Formulation | FR50.14 | FR50.15 | FR50.16 | FR50.17 |
|---|---|---|---|---|
| **pH** | 4.31 | 4.59 | 4.62 | 2.66 |
| $SD_{pH}$ | 0.01 | 0.01 | 0.02 | 0.07 |
| $CV_{pH}$ [%] | 0.23 | 0.22 | 0.43 | 2.63 |
| **Conductivity [$\mu$S/mm]** | 1.7 | 1.1 | 1.1 | 91.6 |
| SDc | 0.0 | 0.1 | 0.1 | 0.9 |
| CVc [%] | 0.0 | 9.1 | 9.1 | 1.0 |

## 4. Discussion

Figure 1a,b compares pH and conductivity achieved by formulations FR50.0–FR50.5 of Table 1 (results reported in Tables 7 and 8) when performing EN 60754-2 at 950 °C and internal method 3. FR50.0, representing a typical non-flame retarded PVC jacket compound for cables, contains a GCC, which is a grade not actually good as an acid scavenger. Internal method 3 and EN 60754-2 differ slightly in pH and conductivity. In particular, for FR50.0, EN 60754-2 only shows a slightly better smoke acidity compared to internal method 3. The phenomenon has been observed in Ref. [7], probably due to the formation of CaO, which more likely occurs at 950 °C than in the heating conditions of EN 60754-1. However, this effect disappears as particle size decreases and $CaCO_3$ increases its efficiency.

FR50.1 and FR50.2, containing ATH and MDH, respectively an inert and an ineffective acid scavenger, show high and comparable smoke acidity with both methods. Hence, the results obtained for FR50.0, FR50.1, and FR50.2 indicate that internal method 3 and EN 60754-2 work similarly when formulations are free of efficient acid scavengers. Nevertheless, the behavior in FR50.3, FR50.4, and FR50.5 is different. All these formulations contain potent acid scavengers at high temperatures that act in the condensed phase. In this case, the differences between the two heating regimes are significant, with EN 60754-2 showing rather higher smoke acidity than internal method 3.

Figure 2a,b compares the pH and conductivity achieved by formulations FR50.6–FR50.9 of Table 2 (results reported in Tables 10 and 11 when performing internal method 3 and EN 60754-2). In this case, the measurements concern the effect on acidity from AS-1B and AS-6B, which are potent acid scavengers at high temperatures, in combination with synthetic MDH. The compounds have high pH and low conductivity, and internal method 3 clearly shows low acidity, confirming the behavior of the samples FR50.3–FR50.5 in Table 1, which also contain potent acid scavengers. It is essential to highlight that as soon as the conductivity reaches values below 10 $\mu$S/mm, obtaining values with less than 5% of the coefficient of variation, as requested by EN 60754-2 (Tables 10 and 11), becomes complex. In fact, the standard has many manual procedures and other sources of errors, exhaustively

explained in [6,7], severely affecting the small values of the conductivity obtained with these formulations.

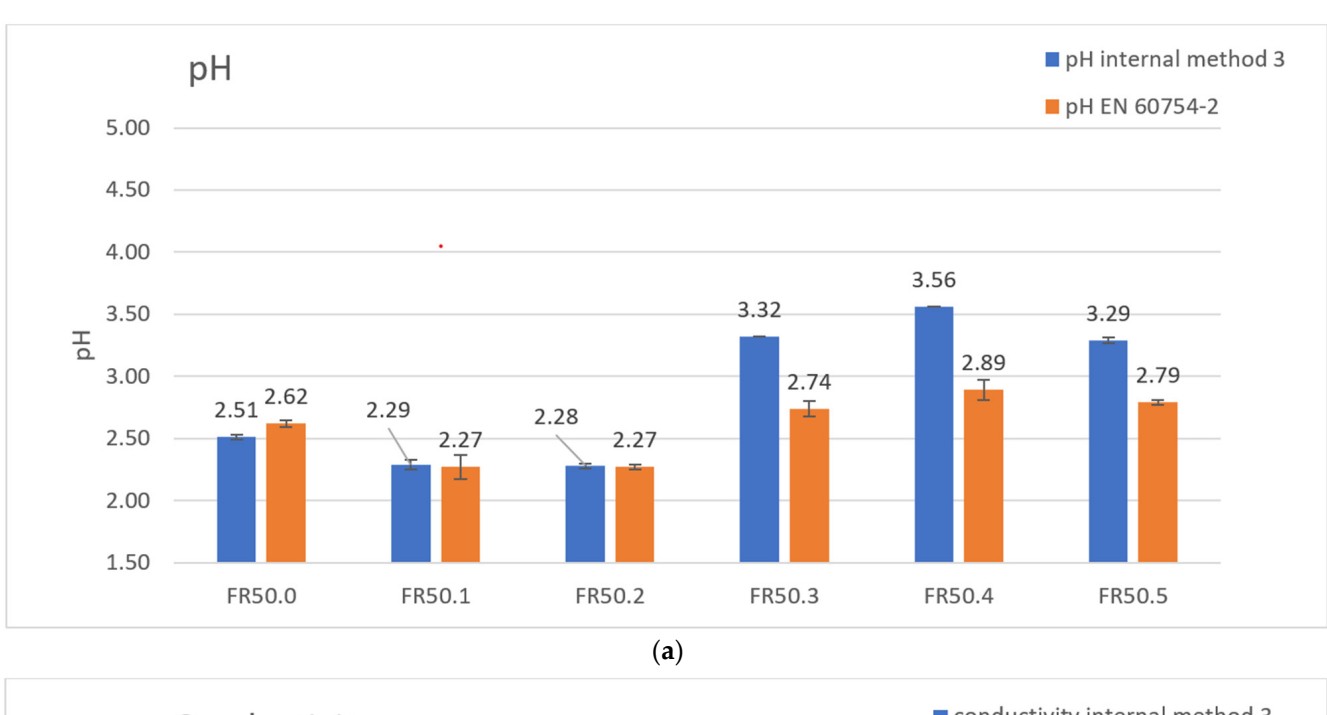

(**a**)

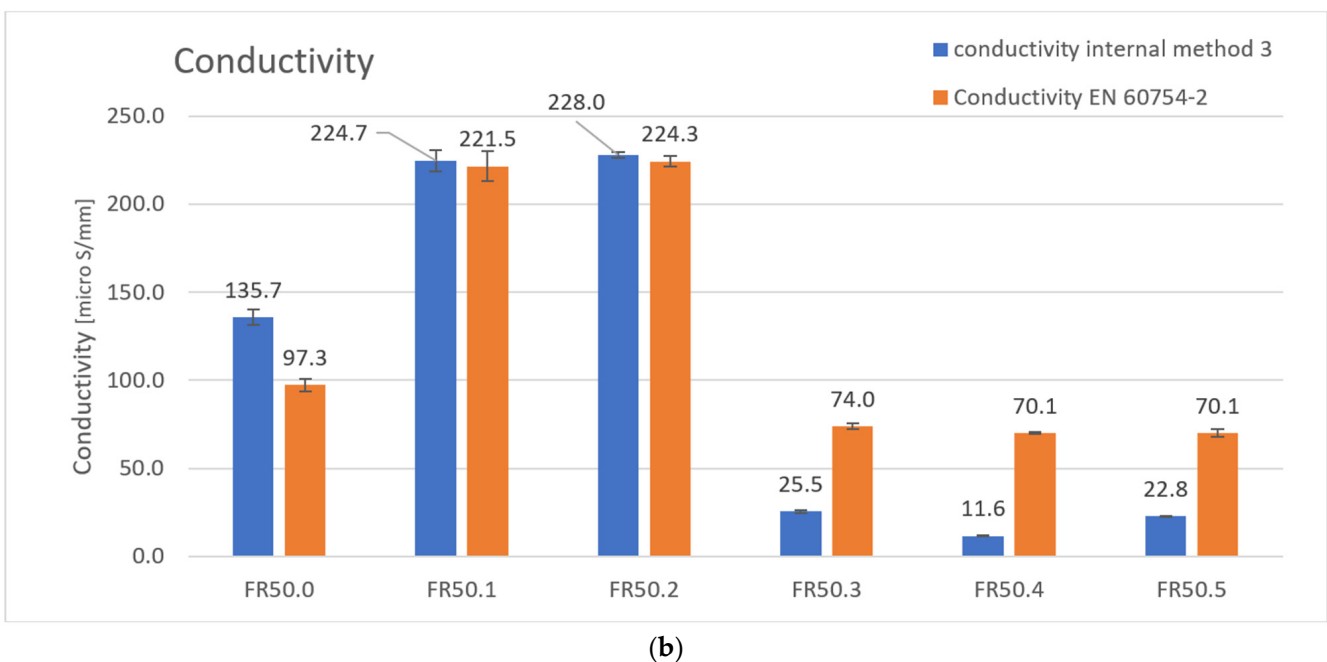

(**b**)

**Figure 1.** Comparison of pH (**a**) and conductivity (**b**) of formulations FR50.0–FR50.5 measured with internal method 3 (blue bars) and EN 60754-2 (orange bars). SD is reported. FR50.0–FR50.2 without efficient acid scavengers, FR50.3–FR50.5 with efficient acid scavengers.

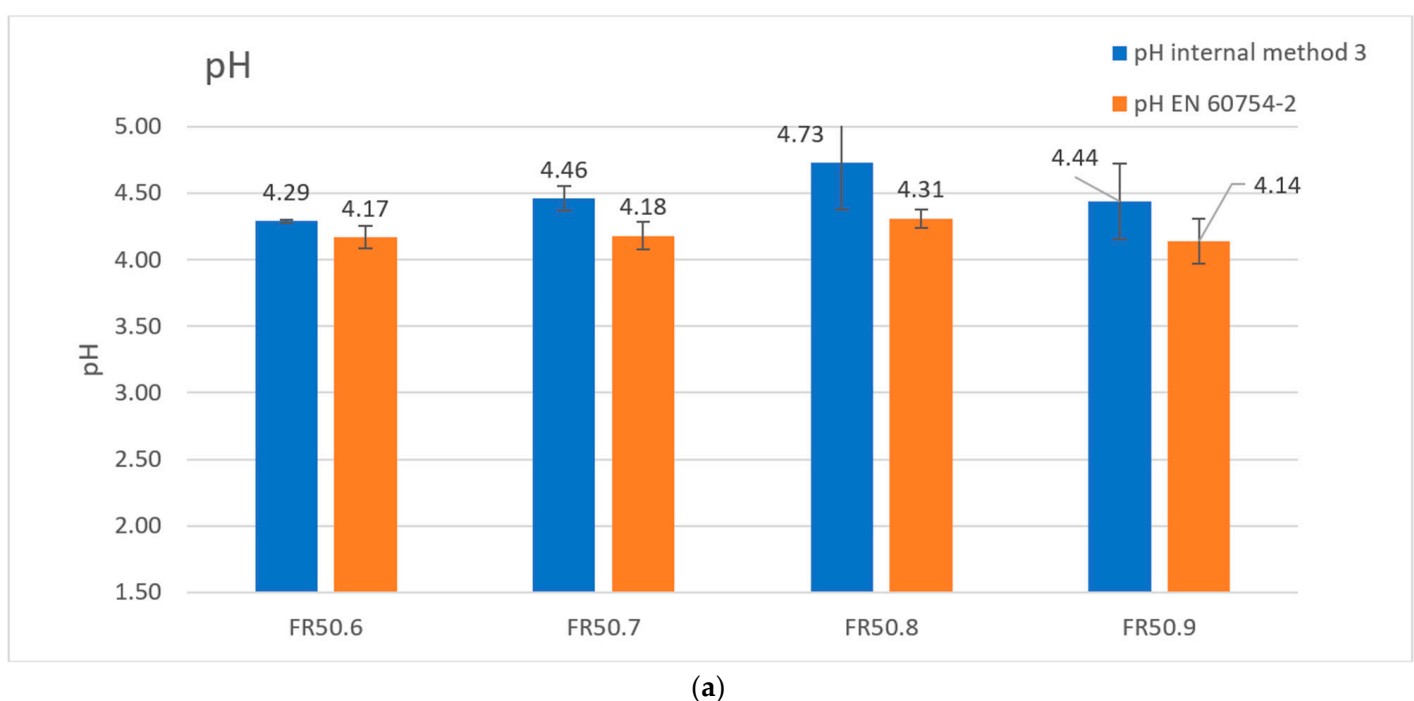

(**a**)

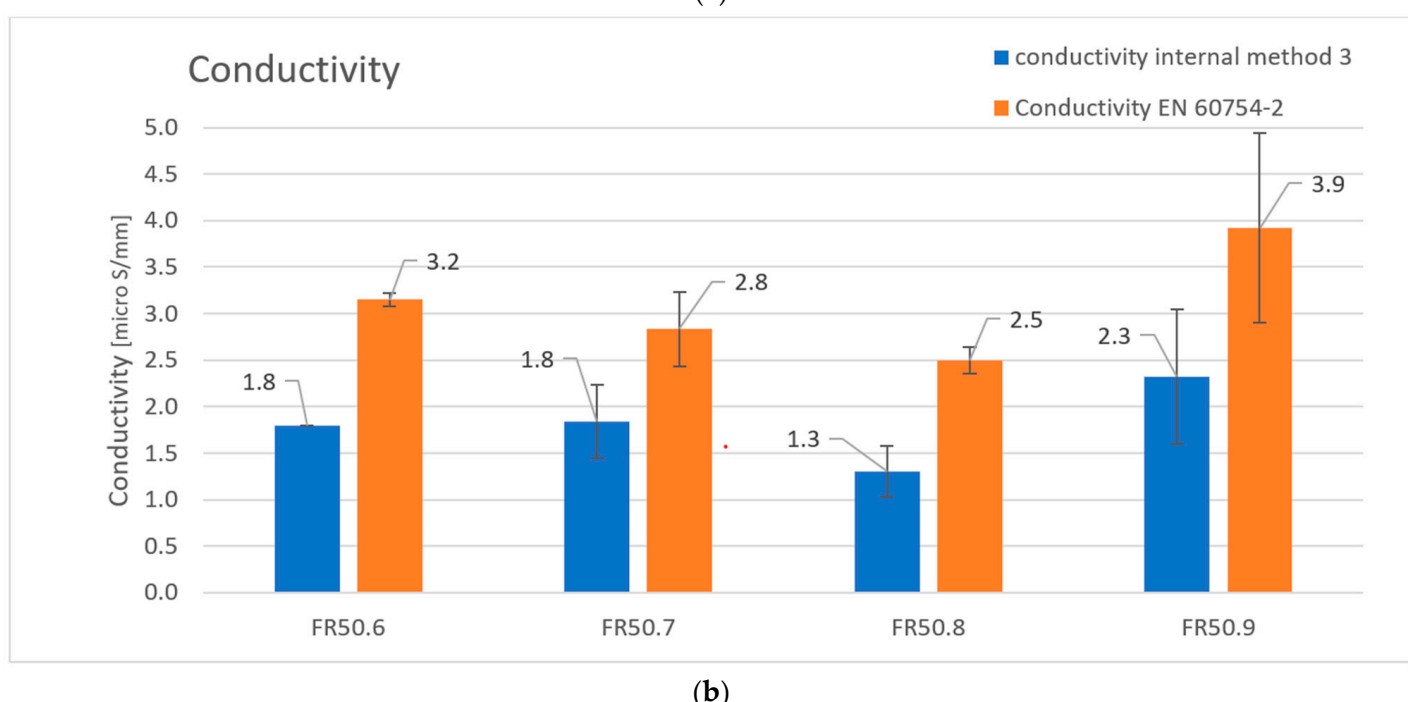

(**b**)

**Figure 2.** Comparison of pH (**a**) and conductivity (**b**) of formulations FR50.6–FR50.9 measured with internal method 3 (blue bars) and EN 60754-2 (orange bars). SD is reported.

Figure 3a,b compares the pH and conductivity achieved by formulations FR50.10–FR50.13 of Table 3 (results reported in Tables 12 and 13), performing internal method 3 and EN 60754-2 at 950 °C. In this set of formulations, UPCC and AS-0B, potent acid scavengers at high temperatures, are tested in combination with synthetic MDH, Kisuma 5A, and fumed silica. All formulations show extremely low acidity, reflecting the synergistic effect of UPCC and AS-0B with MDH in multiple-step reactions, as described in [6,7]. AS-0B behaves better than UPCC. The use of fume silica, aiming to help the dispersion, is unsuccessful.

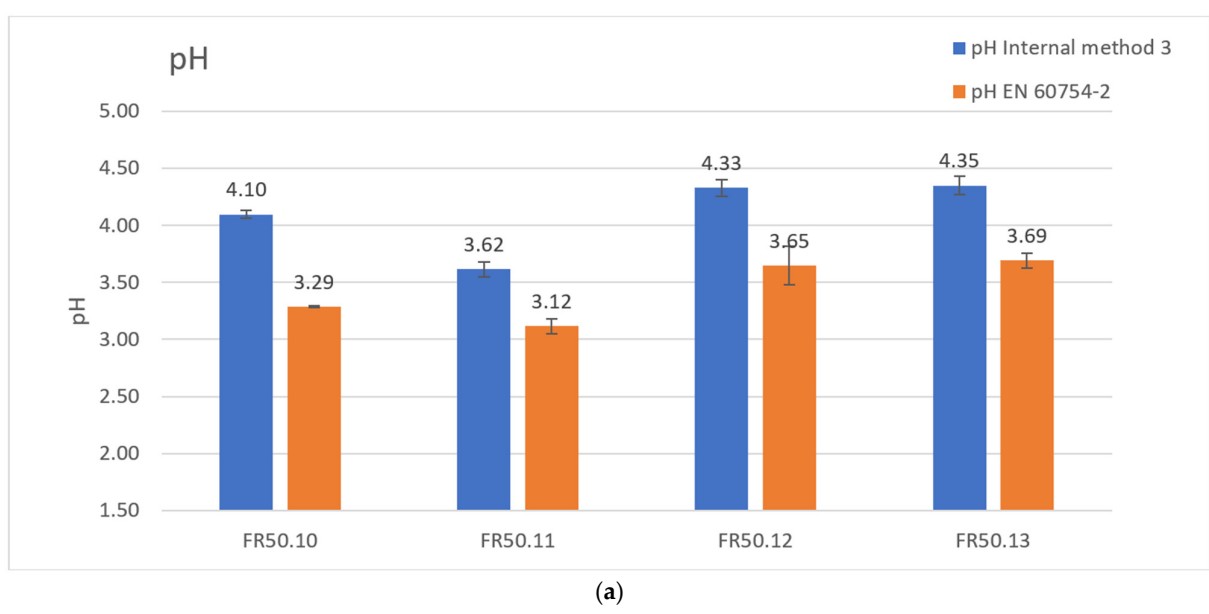

(**a**)

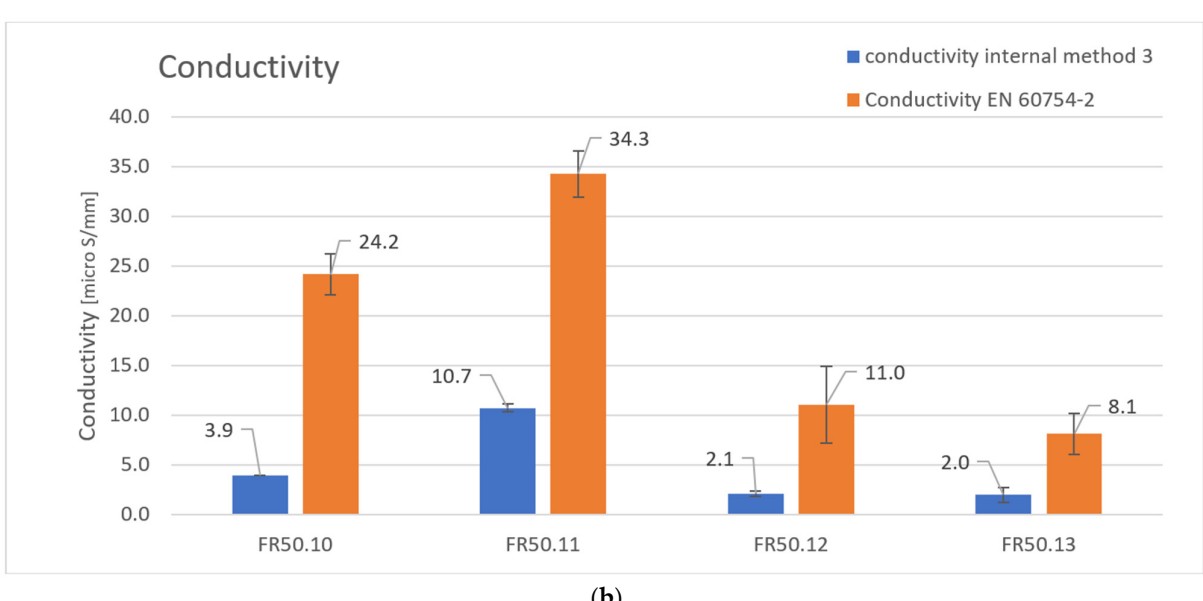

(**b**)

**Figure 3.** Comparison of pH (**a**) and conductivity (**b**) of formulations FR50.10–FR50.13 measured with internal method 3 (blue bars) and EN 60754-2 (orange bars). SD is reported.

As for the formulations of the first two series (Tables 1 and 2), internal method 3 and EN 60754-2 again show significant differences due to the different temperature regimes and final temperatures, with EN 60754-2 measuring higher acidity than internal method 3.

Figure 4a,b compares the pH and conductivity achieved by formulations FR50.14–FR50.17 of Table 4 (results reported in Tables 14 and 15), performing internal method 3 and EN 60754-2 at 950 °C. In this series, AS-0B, AS-1B, and AS-6B, potent acid scavengers at high temperatures, are tested with milled brucite (FR50.14, FR50.15, and FR50.16). As with synthetic MDH, with brucite, the smoke acidity is also low, suggesting its synergistic effect between AS-0B, AS-1B, and AS-6B. Again, the internal method 3 and EN 60754-2 differences are considerable. On the other hand, FR50.17 is a typical CPR jacket compound used for matching the classification Cca s3 d1 a3 in PVC cables. Figure 4a,b and Tables 14 and 15 show that the new low-smoke acidity compounds exhibit acidity values of some orders below standard grade compounds for cable currently on the market. In this last case, being

the compound free of potent acid scavengers, internal method 3 and EN 60754-2 give comparable measurements in terms of pH and conductivity.

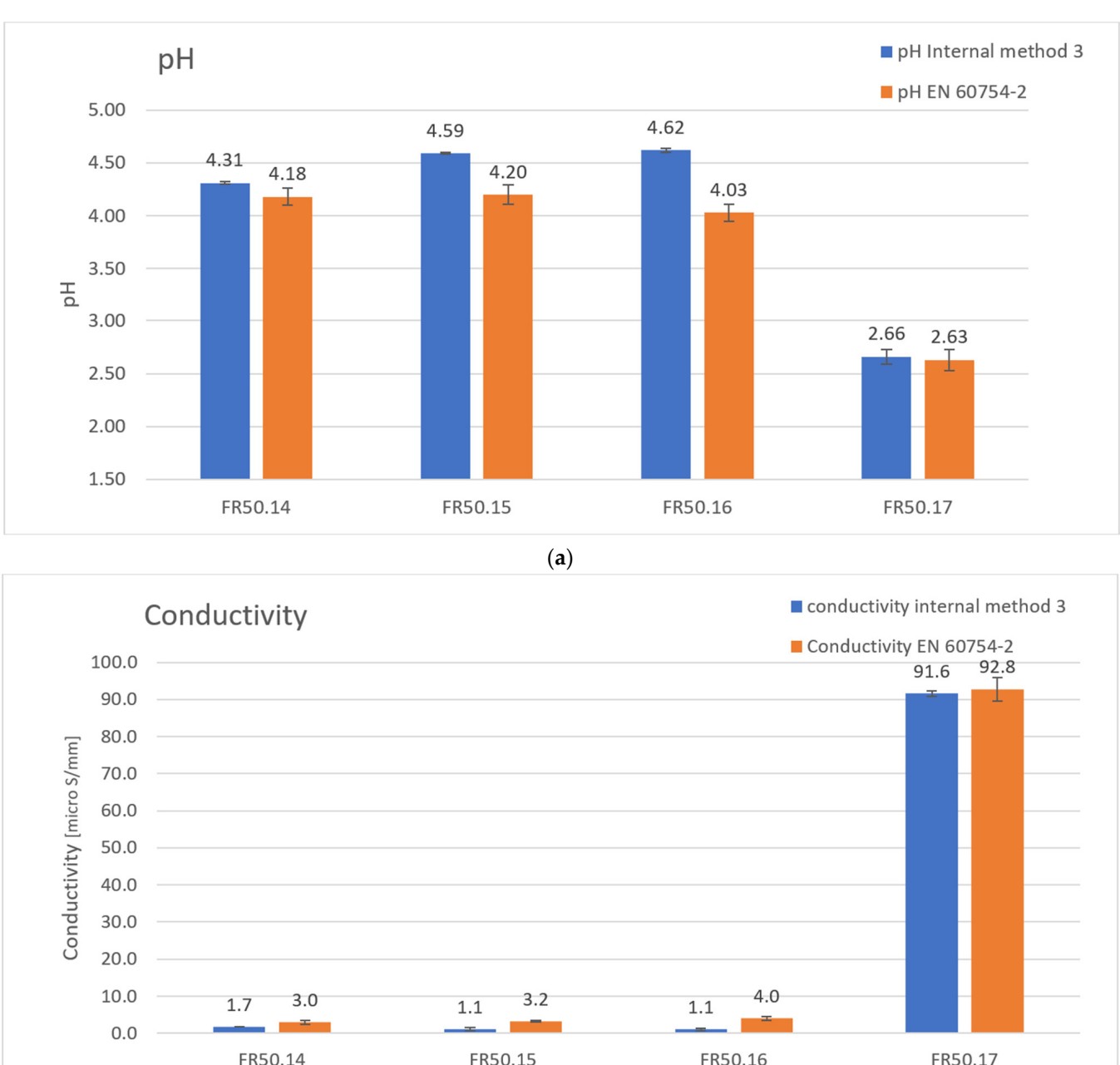

**Figure 4.** Comparison of pH (**a**) and conductivity (**b**) of formulations FR50.14–FR50.17 measured with internal method 3 (blue bars) and EN 60754-2 (orange bars). SD is reported.

Figure 5a,b compares the pH and conductivity (Table 9) achieved by formulations FR50.0–FR50.5 of Table 1, performing internal method 2 and EN 60754-2. The measurements have been performed in isothermal conditions, respectively, at 500 °C and 950 °C and clearly show that applying a lower temperature of 500 °C, in the presence of effective acid scavengers, the HCl in the gas phase is highly reduced (see FR50.3–FR50.5). The behavior of FR50.0 in isothermal conditions at 500 °C resembles that observed with the internal method 3. The formation of CaO at 500 °C is unlike, and the efficiency at 950 °C is higher.

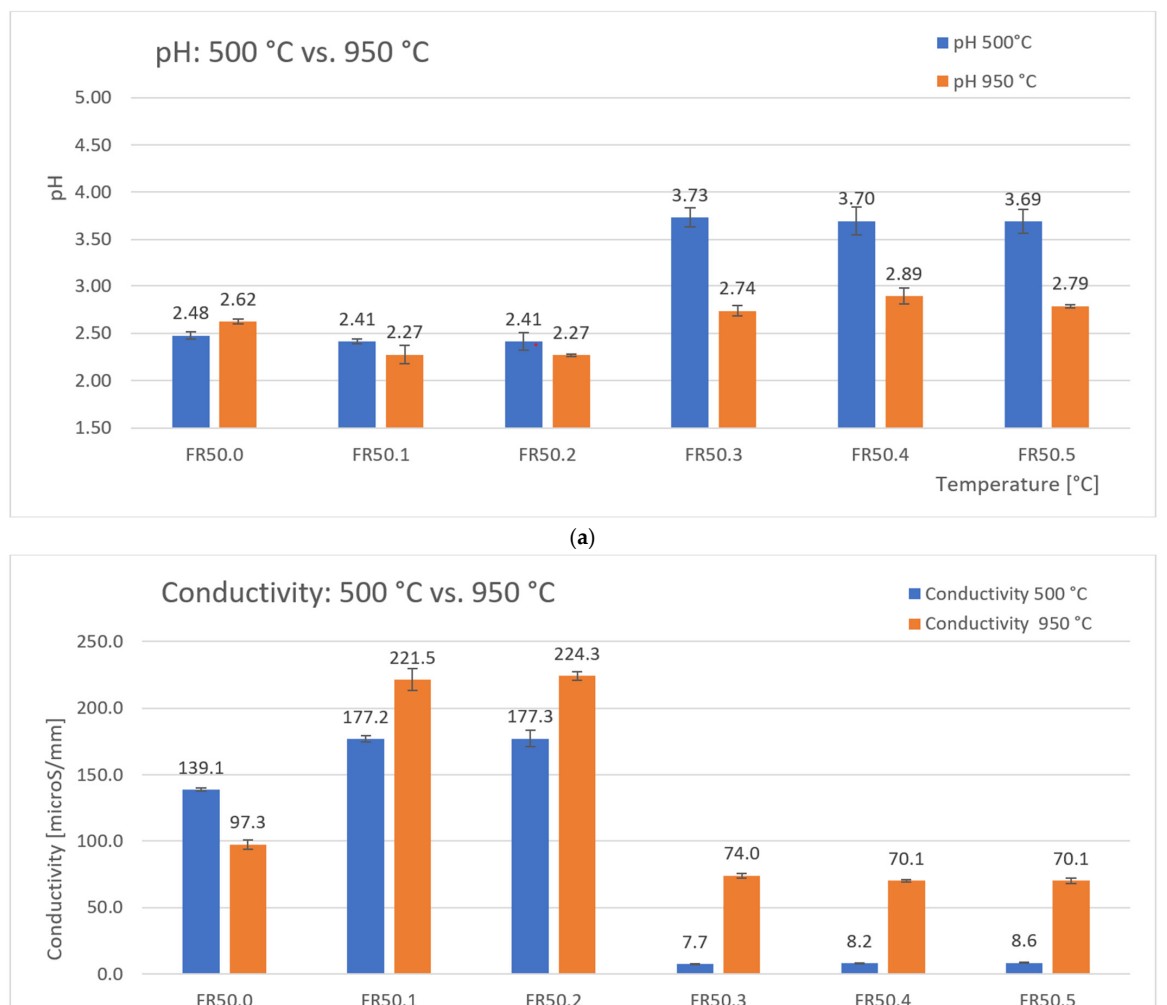

**Figure 5.** Comparison of pH (**a**) and conductivity (**b**) of formulations FR50.0–FR50.5 measured with internal method 2 at 500 °C (blue bars) and EN 60754-2 (orange bars) at 950 °C. SD is reported.

All these data clearly indicate that if a powerful acid scavenger has time to react with HCl, as it happens at lower temperatures (internal method 2, isothermal) and lower temperatures with a slower heating regime (internal method 3), it can trap HCl in the condensed phase with higher efficiencies. On the other hand, the higher temperatures and fast heating regimes of EN 60754-2 hinder its action: the acid scavenger at high temperatures cannot compete with the rapid HCl evolution during PVC compound combustion, and HCl escapes quickly into the gas phase, decreasing the pH and increasing the conductivity of the solutions in the bubblers. In the absence of effective acid scavengers, both standards show comparable values.

The interference of heating regimes and final temperatures in tube furnace tests for determining HCl was well explained in [24] in 1986. Here Chandler, Hirschler, and Smith highlighted that humidity, soot formation, dimensions of the combustion boat, and temperature regimes could affect the results of the method. In the paper, the acid scavenger in PVC compounds was $CaCO_3$, but no information regarding its particle size was given. Isothermal conditions at 650 °C, 700 °C, 800 °C and 950 °C were applied, showing an increase in the emission of HCl as temperature increases. The scavenging efficiency in isothermal was also lower than performing a temperature gradient of 10 °C per minute. All these aspects confirmed that $CaCO_3$, as an acid scavenger, suffers from high temperatures and fast heating rates. They justified the behavior with these specific phrases.

*"It is clear that the higher the temperature at which the tube furnace test is carried out, the higher the HC1 emission will be"*.

*"The lower efficiency during isothermal runs (after 3 weight loss stages) than during gradual heating runs….., coupled with the fact that there is a significantly larger weight loss in the first stage of the isothermal runs, indicates that there is a much greater likelihood of HCl being emitted before it has had the opportunity of reacting with the filler"*.

The acid scavengers used in our paper, currently used such as UPCC or novel such as AS-XB series, confirm the same behavior with a collapse of their efficiencies as temperature increases and when the heating regime accelerates.

## 5. Conclusions

Acidity is the weak point of PVC, and currently, no PVC cables can meet classes $a_1$ or $a_2$, performing EN 60754-2. For this reason, the research on novel acid scavengers capable of decreasing acidity is crucial. In this context, exploring different heating regimes in EN 60754-2 in the presence and absence of acid scavengers is also interesting, particularly when performed with the milder heating regime of EN 60754-1 and in isothermal conditions below flashover (500 °C).

The paper actually highlighted that when an acid scavenger, acting in the condensed phase at high temperatures, is added to the PVC compound, EN 60754-2 performed at 950 °C assesses higher smoke acidity than internal method 3, with a heating regime up to 800 °C. On the contrary, if acid scavengers are absent, inert, or ineffective, both tests show comparable acidity values. Depending on the acid scavengers in the formulations, the gap between EN 60754-2 and internal 3 can become significant. For example, in Figure 1b, FR50.3 gives a value about 3 times lower in conductivity when the gradual heating regime of EN 60754-1 is set, becoming even about 10 times less when EN 60754-2 is run isothermally at 500 °C, using internal method 2 (Figure 5b). This gap is evident for an acid scavenger, such as UPCC, commonly used in PVC compounds for cables to reduce the effluents' acidity in many standards out of the scope of CPR and also for the novel acid scavengers (AS-XB series) studied in this paper in several kinds of formulations.

This behavior confirms what [7] reports, where the efficiencies of some acid scavengers were measured in isothermal conditions at different temperatures. What speeds up the evolution of HCl, such as high temperatures and quick heating regimes, hinders the action of the acid scavengers during PVC compound combustion. Therefore, acid scavengers cannot trap HCl released quickly in the gas phase, increasing the effluents' acidity. Thus, the higher the temperature or faster the heating regime, the quicker the evolution and lower the probability of acid scavenger trapping HCl, as shown also in [24,25].

It must be highlighted that room fires can have different stages with different temperatures and heat flows. Temperatures can rise from 300 °C to 600 °C in the ignition and developing fire stages, capable of reaching from 650 °C up to 1100 °C in the fully developed stage [9,11]. As emerged in this paper and [7], the temperatures and heating regime of EN 60754-2 obliterate entirely the action of the powerful HCl scavengers in low-smoke acidity compounds for cables. Acid scavengers, on the other hand, work efficiently (even up to 10 times better) when a heating regime or pre-flash-over temperatures are used.

In our opinion, the implications of these results and the considerations on room fire temperatures show how EN 60754-2 is weak in its indirect assessment of acidity. It is probably a useless device to foresee if the material of an item can be a real problem in terms of its capability of releasing HCl in the gas phase. That is not only because in real fire scenarios, HCl decays, generating less acidity than expected, and travels only a short distance from where the fire originated [14]. But also because EN 60754-2 assesses the acidity at typical temperatures of fully developed fires, and a different heating regime should probably be adopted. Full-scale fire tests comparing the evolution of HCl of PVC cables of different classes could definitely clarify this aspect.

**Supplementary Materials:** The following supporting information can be downloaded at: https://www.mdpi.com/article/10.3390/fire6080326/s1, Table S1: commercial additives. Table S2: test apparatuses. Table S3: standards. Section S2: sample preparation. Section S3: test procedures.

**Author Contributions:** Conceptualization, G.S.; methodology, G.S., F.D., I.B., M.P. and C.B.; writing—original draft preparation, G.S.; writing—review and editing, G.S., F.D., I.B., M.P. and C.B. All authors have read and agreed to the published version of the manuscript.

**Funding:** This research received no external funding.

**Institutional Review Board Statement:** Not applicable.

**Informed Consent Statement:** Not applicable.

**Data Availability Statement:** Not applicable.

**Acknowledgments:** The authors want to acknowledge Carlo Ciotti, Emma Sarti, all PVC Forum Italia, and the PVC4cables staff.

**Conflicts of Interest:** The authors declare that there is no conflict of interest regarding the publication of this paper.

## Abbreviations

| | |
|---|---|
| PVC | Poly(vinyl chloride) |
| HCl | Hydrogen chloride |
| EU | European Union |
| CPD | Construction Product Directive |
| CPR | Construction Product Regulation |
| UPCC | Precipitated Calcium Carbonate |
| GCC | Ground Calcium Carbonate |
| Phr | Part per Hundred Resin |
| DINP | Di Iso Nonyl Phthalate |
| ESBO | Epoxidized Soy Bean Oil |
| COS | Calcium Organic Stabilizer |
| DDW | Double Deionized Water |
| M | Mean |
| SD | Standard Deviation |
| CV | Coefficient of variation |

## Appendix A. A Schematic Diagram of the Sample Preparation and Testing Process

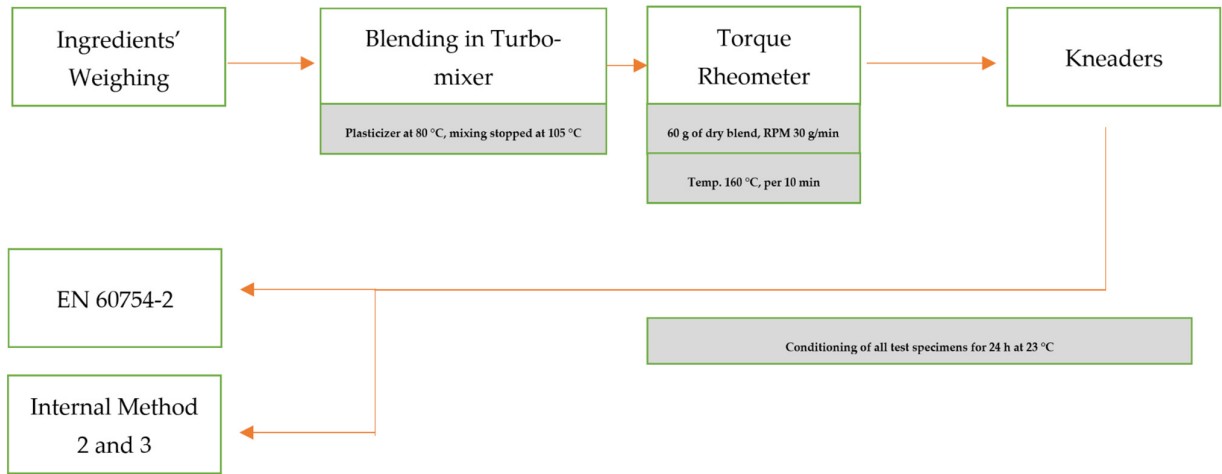

**Figure A1.** A schematic diagram of the sample preparation.

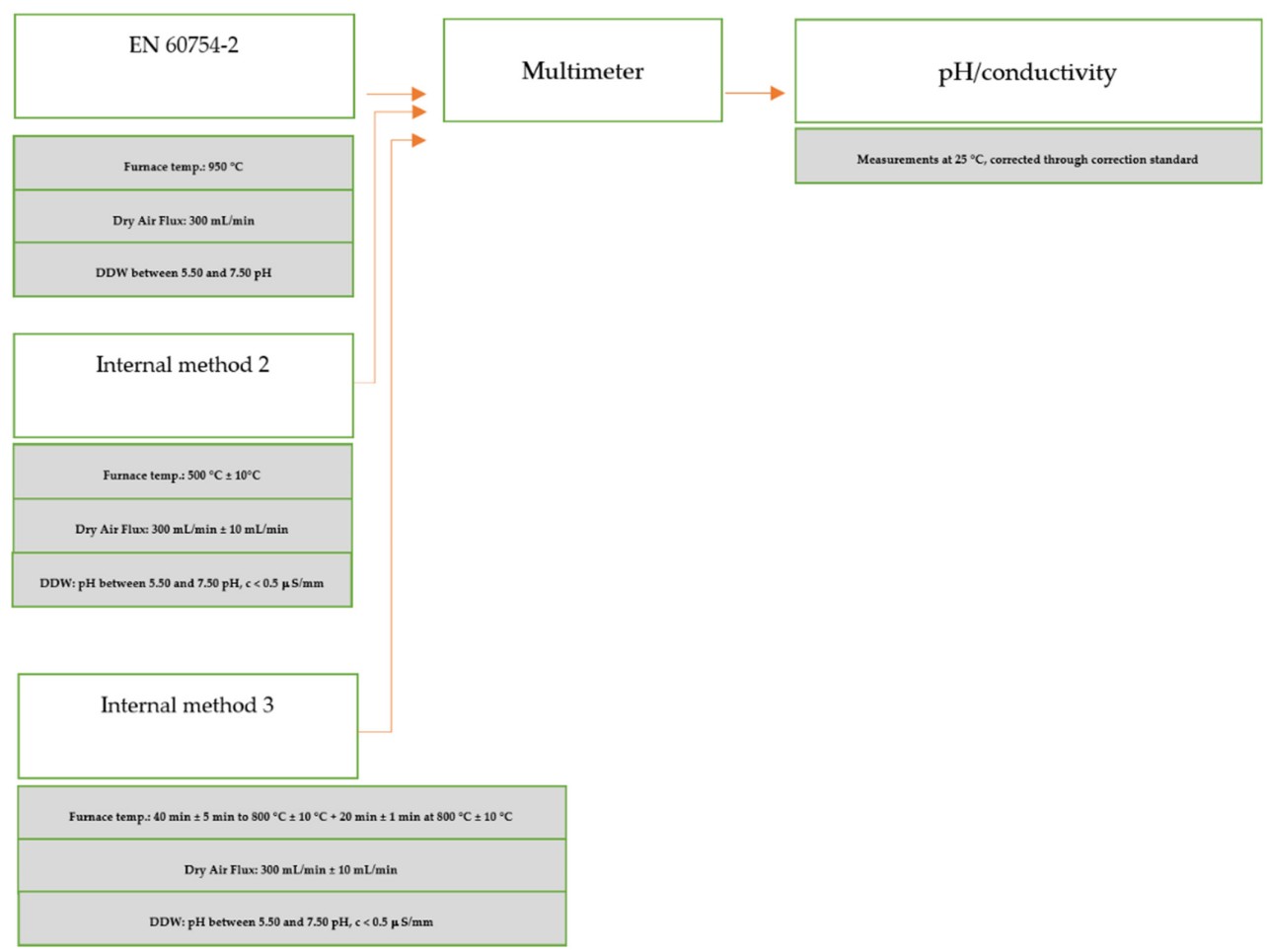

**Figure A2.** A schematic diagram of the testing process and main conditions.

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
