# Peer review of "A New Perspective on Hydrogen Chloride Scavenging at High Temperatures for Reducing the Smoke Acidity of PVC Cables in Fires V: Comparison between EN 60754-1 and EN 60754-2"

_fire, doi:10.3390/fire6080326_

Round 1

Reviewer 1 Report

The concepts and conclusions of this work are reasonable and they highlight a potential problem for manufacturers of wires or cables insulated with PVC (whether as the insulation or the jacket): the test method used to assess acid gas emission negatively affects the formulations that can be used to decrease acid gas emission, especially when compared to other potential test methods or to actual fire scenarios.

It needs to be pointed out that "acidity" is an "additional declaration" and not a regulatory requirement, but the authors are correct in that (a) no other construction product in EN 13501 contains an "acidity" additional declaration and (b) that it is often included in specifications, even if though is not a regulatory requirement. This should be clarified.

The description of "internal method 2" and "internal method 3" needs to be improved. It appears that the difference between "internal method 2" and IEC 60754-2 is simply that the furnace is used isothermally at a lower temperature, but I am not sure that this assumption is correct. 

I wonder whether there is a need to present the same information in tables and in figures. I suspect the figures are unnecessary duplication.

The English could be improved but it is understandable.

Reviewer 2 Report

The comparative experiments were well designed and the results are of high quality. Results show a new perspective on hydrogen chloride scavenging at high temperatures for reducing smoke acidity of PVC cables in fires between EN 60754-1 and EN 60754-2.  Also, the authors made a discussion by comparing different characteristic parameters. My major comments are:

(1)   The abstract should be rewritten. The introduction part of the research background should be appropriately deleted, and a brief description of the experimental method, the main results of the experiment and important views and conclusions should be included.

(2)   In Introduction Section, it needs significant reworking to clarify the situation and the scientific issues on related researches.

(3)   Though the paper is easy to read, and all the results were discussed and obtained based on comparative methods, without further theoretic explanation on the phenome it would definitely limit the meaning of the study.

(4)   It is recommended that room fire experiments can be further developed to obtain the real reducing rate of HCL with samples.

Also, it is suggested that the figures should be re-plotted (especially should use unified coordinate scale). I don’t think the quality of the figures will meet with approval.

Reviewer 3 Report

The manuscript is technically well written. However, some issues are noticed as listed below.

How to perform  Hydrogen Chloride Scavenging at High Temperatures?

Are there any methods discussed in the literature about reducing the Smoke Acidity of PVC Cables in Fires?

For Materials and Methods, provide a schematic diagram.

What is new and novelty of this work?

The sample preparation is to be explained in detail.

How to measure the pH and conductivity of present samples?

Conclusions need to be rewritten in a precise manner.

average

Round 2

Reviewer 2 Report

I think it could be accepted now.